# A novel speech emotion recognition method based on feature construction and ensemble learning

Yi Guo[1], Xuejun Xiong[1]*, Yangcheng Liu[1], Liang Xu[1], Qiong Li[2]

**1** Electrical Engineering and Electronic Information, Xihua University, Chengdu, China, **2** The 52nd Research Institude of China Electronics Technology Group Corporation, Haidian, China

* xlwj_sd@163.com

**Data Availability Statement:** Data cannot be shared publicly because of copyright. Data are available from the Institute of automation, Chinese Academy of Sciences for researchers who meet the criteria for access to confidential data. The data

## Abstract

In the field of Human-Computer Interaction (HCI), speech emotion recognition technology plays an important role. Facing a small number of speech emotion data, a novel speech emotion recognition method based on feature construction and ensemble learning is proposed in this paper. Firstly, the acoustic features are extracted from the speech signal and combined to form different original feature sets. Secondly, based on Light Gradient Boosting Machine (LightGBM) and Sequential Forward Selection (SFS) method, a novel feature selection method named L-SFS is proposed. And then, the softmax regression model is used to learn automatically the weights of the four single weak learners including Support Vector Machine (SVM), K-Nearest Neighbor (KNN), Extreme Gradient Boosting (XGBoost) and LightGBM. Lastly, based on the learned automatically weights and the weighted average probability voting strategy, an ensemble classification model named Sklex is constructed, which integrates the above four single weak learners. In conclusion, the method reflects the effectiveness of feature construction and the superiority and stability of ensemble learning, and gets good speech emotion recognition accuracy.

## 1 Introduction

Language is the main method of human communication. The current field of Human-Computer Interaction is no longer limited to keyboards and screens, and it has been extended to speech interaction, such as Baidu's 'Xiaodu', Microsoft's 'Cortana', Apple's 'Siri' and other smart speech assistant. As the external form of language, speech has rich emotional information, which makes communication more appropriate, smooth and efficient. However, speech interaction of these speech assistants is relatively monotonous without emotion fluctuations, which makes users feeling bored. Now, mining emotional information from the speaker's speech data is a hot research interest in the field of speech signal processing. It is of great significance for boosting the level of Human-Computer Interaction, and has very important value for academic research and practical application.

In recent years, researchers have devoted themselves to the research of speech emotion recognition and have made progress. With the intention of solving the problem of low

underlying the results presented in the study are available from (Address:95 Zhongguancun East Road, Haidian District, Beijing 100190 Room 514, intelligent building, Chinese Language Resource Alliance, URL:http://www.chineseldc.org/, Email: service@chineseldc.org, Tel:86 10 82544772).

**Funding:** This work was supported by Research Funds of Sichuan Provincial Key Laboratory on Intelligent Terminals under grant (SCITLAB-1021) and the National Natural Science Foundation of China under Grant (61973257, 61901394). There was no additional external funding received for this study. The funders had no role in study design, data collection and analysis, decision to publish, or preparation of the manuscript.

**Competing interests:** The authors have declared that no competing interests exist.

classification accuracy caused by single weak learner, literature [1] proposed a boost classification model based on the combination of AdaBoost and ELM. On feature set ISO9, its accuracy was 6% higher than that of single weak learner. Literature [2] proposed a RxK speech emotion recognition model based on Linear Discriminant Analysis (LDA) dimensionality reduction, which gained a 4% higher accuracy than that of single model on the feature set IS13. Insufficient emotional expression is another cause for low classification accuracy. Taking this into account, literature [3] proposed a speech emotion recognition method combining Mel Frequency Cepstrum Coefficient (MFCC) and spectrogram features, thus boosted the classification accuracy that was 29.4% and 23.8% higher than MFCC and spectrogram features respectively. Literature [4] proposed an improved frequency band partitioning OBSI feature extraction method suitable for speech emotion recognition, which was widely used in the field of traditional speech recognition. Compared with the applied MFCC features, the average recognition rate of OBSI and OBSIR features increased by 8.72% and 7.32%, respectively. In order to solve the problem of the low recognition caused by irrelevant and redundant features, literature [5] proposed a fisher feature screening decision tree support vector machine model. Its average recognition accuracy increased by 8%.

Although speech emotion recognition has made progress and development, there is still room for improvement in the recognition accuracy. Inspired by those methods mentioned above problem-solving, this paper proposes a novel speech emotion recognition method based on feature construction and ensemble learning. In this method, for the purpose of boosting the recognition accuracy, this paper explores the speech emotion expression ability of different feature sets. A feature selection method named L-SFS is used to remove the irrelevant and redundant features in high-dimensional data, which boosts the calculation speed and accuracy of model. The combination ability of ensemble learning is used to boost the poor recognition accuracy and instability of single weak learner, which improves the generalization ability of the model. The softmax regression model is used to automatically assign weights to single weak learners, which strengthens the integration ability of ensemble learning, and boosts the recognition accuracy.

## 2 Speech emotion recognition framework

Speech emotion recognition framework is shown in Fig 1, which mainly includes two stages: feature construction and classification model construction.

In the feature construction stage, it includes two steps feature extraction and feature selection. Firstly, the time-domain, spectrogram-domain and cepstrum features of speech signal are extracted. And then, the extracted features are combined into different original feature sets. Finally, the feature set is obtained by selecting the original feature set.

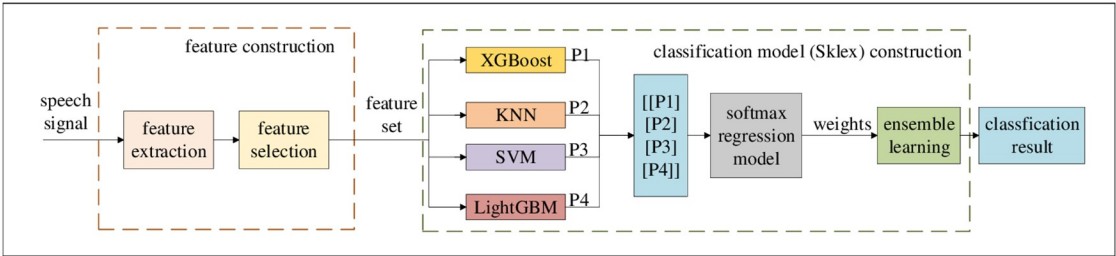

**Fig 1. Speech emotion recognition framework.**

**Table 1. Different features and its global statistics.**

| Serial number | Feature names | Statistical indicators |
|---|---|---|
| 1–8 | STEN and its first-order difference | mean, variance, maximum and minimum |
| 9 | AVSS | None |
| 10–17 | PFCV and its first-order difference | mean, variance, maximum and minimum |
| 18–47 | FFCY and its first-order difference | mean, variance, maximum, minimum, and median |
| 48–125 | 13 dimensional MFCC | mean, variance, maximum, minimum, median and sum |
| 126–203 | 13 dimensional GFCC | |
| 203–281 | 13 dimensional BFCC | |
| 281–359 | 13 dimensional NGCC | |
| 360–437 | 13 dimensional MSRCC | |
| 438–515 | 13 dimensional PSRCC | |
| 516–593 | 13 dimensional LFCC | |

In the classification model construction stage, the feature set is used to train the four single weak learners including XGBoost, KNN, SVM and LightGBM. Then, the output of the above four learners is combined into the input of softmax regression model, and the weight of each learner is learned automatically. Based on the learned weights and the weighted average probability voting strategy, the above four single weak learners are integrated and named Sklex. Finally, the feature set is classified by Sklex.

## 3 Feature construction

### 3.1 Feature extraction

Feature extraction is the key step to speech emotion recognition [6]. The time-domain and cepstrum features of speech signal are extracted in this paper. The time-domain features include Short-Time Energy (STEN), Pitch Frequency (PFCY), Formant Frequency (FFCY) and Average Speech Speed (AVSS) [7]. The cepstrum features include MFCC, Gamma Frequency Cepstrum Coefficient (GFCC) [8], Barker Frequency Cepstrum Coefficient (BFCC) [9], Normalized Gamma Chirped Cepstrum Coefficient (NGCC) [10], Amplitude-based Spectrum Root Cepstral Coefficient (MSRCC), Phase-based Spectrum Root Cepstral Coefficient (PSRCC) [11] and Linear Frequency Cepstrum Coefficient (LFCC) [12].

After extracted features from speech signal, the global statistics of the features including mean, variance, maximum, minimum, median and sum, are used as the input of classification model, as shown in Table 1.

Feature combination, a method is combining multiply different features, not only can improve the emotion expression ability of different features, but also eliminate the redundant information caused by the correlation between different features [13]. Parallel connection is adopted to combine different feature sets, which is essential to expand the feature dimension. According to the serial number and features in Table 1, the features are combined into ten different feature sets and named, as shown in Table 2. Where, STTF is a time-domain feature set. CESP is a cesptrum feature set. STTFCESP combines STTF and CESP. And others are single cesptrum feature set.

### 3.2 Feature selection

In the application of machine learning, the dimension of feature set tends to be high, and irrelevant and redundant features always exist. Feature selection can eliminate the features, so as to

**Table 2. Different feature sets.**

| Serial number in Table 1 | Feature set name | Serial number in Table 1 | Feature set name |
|---|---|---|---|
| 1–47 | STTF | 282–359 | NGCCs |
| 48–125 | MFCCs | 360–437 | MSRCCs |
| 126–203 | GFCCs | 438–515 | PSRCCs |
| 204–281 | BFCCs | 516–593 | LFCCs |
| 48–593 | CESP | 1–593 | STTFCESP |

reduce feature dimension, improve accuracy of model and decrease the running time of program. For the purpose of making the constructed model better, this paper researches numerous feature selection methods, which aims to create a feature subset possessing extraordinary expression power of speech emotion from original feature set [14].

Feature selection process generally includes four steps namely generation, evaluation function, stop criterion and verification [15]. The feature subset is searched in the generation process and feed into evaluation function that is a criterion to evaluate the quality of a feature subset. The stop criterion is related to the evaluation function, and generally a threshold. When the value of the evaluation function reaches this threshold, the search will be stopped. The verification process verifies the effectiveness of the selected feature subset. In view of this process, this paper proposes a feature selection method based on LightGBM and SFS, and named L-SFS, its specific steps are as follows.

1. Sort the feature set. The importance degree of features is learned through LightGBM, and the features are sorted in descending order by the importance degree.

2. Initialize settings. Set an empty feature set $X$, and a parameter $crr0$ with classification accuracy of 0.

3. Add a feature $x$ to $X$. In the light of the order of feature, the first feature $x$ is added to $X$.

4. Get the new learner accuracy $crr1$. The feature set $X$ is classified by SVM to get $crr1$.

5. Select feature. If $crr1 >= crr0$, the feature $x$ is helpful to improve the accuracy of learner, then the feature $x$ is retained in $X$ and the value of $crr0$ is updated by the value of $crr1$. Else, it means that the feature $x$ is not conducive to boosting the accuracy of the learner, then feature $x$ is deleted from $X$.

6. Remove $x$ from the feature set.

7. Repeat (3)-(6), and stop after traversing all features.

8. SVM is used to verify the selected feature set.

# 4 Classification model construction

## 4.1 Voting strategy about ensemble learning

There are two main voting strategy about ensemble learning [16]: the majority voting strategy and the weighted average probability voting strategy.

In the majority voting strategy, each learner vote for categories. The category that gets the most votes is the final prediction category. If category 1 and category 2 get four votes respectively, and category 1 reaches four votes earlier than category 2, category 1 is the final

prediction category. The strategy is more likely to be affected by the same number of votes and randomness, resulting in inaccurate final prediction results. In addition, the strategy require that each learner has the same voting ability, but in real experiments, the voting ability of each learner is difficult to keep consistent, which will affect the final prediction results.

In the weighted average probability voting strategy, specific weights can be assigned to each learner. After providing the weights, the predicted category probabilities for each learner are obtained, multiplied by the weights, and averaged [17]. The final prediction category is derived from the category label with the highest average probability.

Compared with the majority voting strategy, the weighted average probability voting strategy reflects the inconsistency of the learner, can emphasize the performance of the learner, and help improve the accuracy of model. And there are few cases with the same probability, so it barely affects the classification results. According to the above comparison, this paper chooses the weighted average probability voting strategy. Building on the four single weak learners, SVM [18], KNN [19], XGBoost [20], and LightGBM [21], the classication model is constructed. The description of the classification model based on weighted average probability voting strategy is as follows.

1. SVM, KNN, XGBoost and LightGBM are defined as No.1 to No.4 learners respectively.

2. The weights of the four learners is $W$.

$$W = [w_1, w_2, w_3, w_4]_{1 \times 4} \tag{1}$$

3. The selected feature set of the speech signal to be recognized are feed to learners No. 1 to No. 4 respectively, and the output probability of each emotion category predicted by the No. $i$ learners is $P_i$.

$$P_i = [P_{i1}, \ldots, P_{ij}, \ldots, P_{i6}]_{1 \times 6} \tag{2}$$

where, $i$ is the learner number, $i \in [1, 2, 3, 4]$. $P_{ij}$ is the output probability of $i$ learner for the $j$ category, $j \in [1, 2, 3, 4, 5, 6]$.

4. According to the weights and the probability of prediction category, the integrated output probability $y_j$ of the $j$ category is obtained, as it shown in Eq 3.

$$y_j = w_1 \times P_{1j} + w_2 \times P_{2j} + w_3 \times P_{3j} + w_4 \times P_{4j} \tag{3}$$

The probability of each prediction category is $Y$.

$$Y = [y_1, y_2, y_3, y_4, y_5, y_6]_{1 \times 6} \tag{4}$$

5. The prediction category corresponding to the maximum value in $Y$ is selected as the final prediction category.

## 4.2 Learn automatically weights

In the weighted average probability voting strategy, the weights assigned to the four single weak learners are random and manual, which possess highly arbitrary and unscientific [22]. Aiming at this problem, this paper uses the softmax regression model to learn automatically weights.

The softmax regression model is a linear superposition model of input features and weights, and the number of output values is equal to the number of categories, so as to achieve the goal of classification. In the ensemble learning based on the weighted average probability voting strategy, the classification results of each single weak learner are obtained firstly, and then the classification results are superimposed by weights, and finally the emotion classification results are obtained. In this process, the classification results of each weak learner can be combined into the input features of softmax regression model, and the weights in the softmax regression model can be regarded as the weights which assigned to each learner. So, when the softmax regression model gets optimal, the weights obtained can also obtain the optimal effect in ensemble learning. Consequently, the learned automatically weights can be realized. To illustrate this process, a single vector calculation process sample is described as follows.

1. Set the weight $W$ of softmax regression model.

$$W = [w_1, w_2, w_3, w_4]_{1 \times 4} \tag{5}$$

Where, $W$ is the weight assigned to the four signle weak learners.

2. Set the features $x^{(d)}$ of sample $d$.

$$x^{(d)} = \begin{bmatrix} P_{11} & P_{12} & P_{13} & P_{14} & P_{15} & P_{16} \\ P_{21} & P_{22} & P_{23} & P_{24} & P_{25} & P_{26} \\ P_{31} & P_{32} & P_{33} & P_{34} & P_{35} & P_{36} \\ P_{41} & P_{42} & P_{43} & P_{44} & P_{45} & P_{46} \end{bmatrix}_{4 \times 6} \tag{6}$$

3. Get the output $O^{(d)}$ of the output layer.

$$O^{(d)} = W x^{(d)} = [O_1^{(d)}, \ldots, O_j^{(d)}, \ldots, O_6^{(d)}]_{1 \times 6} \tag{7}$$

Where, $O_j^{(d)}$ is the output of the output layer to the $j$ prediction category of sample $d$, $j \in [1, 2, 3, 4, 5, 6]$.

4. Softmax operation is performed on $O^{(d)}$ to obtain the probability $Y_j$ of prediction category $j$ as follows:

$$y_j = \frac{\exp\left(O_j^{(d)}\right)}{\sum_{j=1}^{6} \exp\left(O_j^{(d)}\right)} \tag{8}$$

Where, $\exp(^*)$ is the exponential operation.

## 4.3 Sklex method description

Sklex is an ensemble method based on the learned automatically weights and the weighted average probability voting strategy. Firstly, the dimension of feature set is reduced, and then four popular and efficient classification methods namely SVM, KNN, LightGBM and XGBoost are used to construct Sklex. For explicit description of the method, the following definition is given firstly, and then the description is shown in Table 3.

**Table 3. Sklex algorithm.**

| Input: | The training data set is $T = \{(x_i, y_i)\}$, $i = 1, 2, \ldots\ldots, n$, $x_i \in R^n$, $y_i \in \{0, 1, \ldots\ldots, k\}$, $i, k \in N^+$ |
|---|---|
| | The learners $H_1$, $H_2$, $H_3$ and $H_4$ |
| Process: | Reduce the dimensionality of the training data set $T$ to get $T'$ |
| | for $i = 1, 2, \ldots, n$ do |
| | for $j = 1, 2, 3, 4$ do |
| | $P_1 = [P_{11} \, P_{12} \ldots P_{1k}]_{1 \times k} = H_1(T')$, |
| | where $P_1$ is the prediction class probability vector of SVM. |
| | $P_2 = [P_{21} \, P_{22} \ldots P_{2k}]_{1 \times k} = H_2(T')$, |
| | where $P_2$ is the prediction class probability vector of KNN. |
| | $P_3 = [P_{31} \, P_{32} \ldots P_{3k}]_{1 \times k} = H_3(T')$, |
| | where $P_3$ is the prediction class probability vector of XGBoost. |
| | $P_4 = [P_{41} \, P_{42} \ldots P_{4k}]_{1 \times k} = H_4(T')$, |
| | where $P_4$ is the prediction class probability vector of LightGBM. |
| | end for |
| | $P_i = [P_1 \, P_2 \, P_3 \, P_4]^T$, |
| | where $P_i$ is a matrix which is formed by the prediction probability vectors $P_1 \, P_2 \, P_3$ and $P_4$. |
| | end for |
| | Construct training set $D = \{(P_i, y_i)\}$, $i = 1, 2, \ldots\ldots, n$, $x_i \in R^n$, $y_i \in \{0, 1, \ldots\ldots, k\}$, $i, k \in N^+$, |
| | where $P_i$ is the feature and |
| | $y_i$ is the sample label. |
| | Set the weight of the softmax regression model to $W = [w_1 \, w_2 \, w_3 \, w_4]_{1 \times 4}$. |
| | Input the data set $D$ into the softmax regression model, and learn the weight $W$. |
| | for $i = 1, 2, \ldots, n$ do |
| | $y' = \frac{1}{4}(W \times P_i)$, |
| | where $y'$ is the predicted probability of the ensemble model for sample $i$. |
| | $predict_i = argmax(y')$, |
| | where $predic_i$ is the prediction category which is selected by the largest sum of probabilities.s |
| | end for |
| Output: | Predict category $Predict = \{predict_1, predict_2, \ldots, predict_n\}$. |

Def1. Def1 The training data set is $T = \{(x_i, y_i)\}$, $i = 1, 2, \ldots\ldots, n$, $x_i \in R^n$, $y_i \in \{0, 1, \ldots\ldots, k\}$, $i, k \in N^+$, where, $x_i$ is the feature vector, $y_i$ is the sample label, $n$ is the number of samples and $k$ is the number of categories.

Def2. The learners SVM, KNN, XGBoost, and LightGBM are represented by $H_1$, $H_2$, $H_3$ and $H_4$.

Def3. Training set $T' = \{(x'_i, y_i)\}$, $i = 1, 2, \ldots\ldots, n$, $x'_i \in R^n$, $y_i \in \{0, 1, \ldots\ldots, k\}$, $i, k \in N^+$ after dimensionality reduction, where $x'_i$ is the feature vector after $x_i$ projection.

Def4. The predicted value $y'$ of the ensemble model, the weighted average probability value of the ensemble model for each category.

Def5. Predicted category $Predict$, a total of $k$ categories, the category corresponding to the maximum value of the predicted value $y'$.

In this classification method, the learned automatically weights is a highlight, which efficiently improve the speech emotion recognition accuracy. Besides, facing the CASIA database, the ensemble method gets higher recognition accuracy and stability than single weak learner.

## 5 Experimental results and discussions

The experiment aims to test the speech emotion expression ability of different feature sets, to verify the effectiveness of feature selection method named L-SFS, to verify the effectiveness of

learned automatically weights, and to test the speech emotion recognition ability of Sklex method. Moreover, the experimental is designed on the basis of the control variate method.

## 5.1 Database

CASIA Chinese emotional corpus was recorded by the Institute of Automation, Chinese Academy of Sciences. It includes four professional speakers and six kinds of emotions: anger, fear, happiness, neutral, sadness and surprise, a total of 7200 different pronunciation. 300 of the corpus are the same text. That is to say the same text given different emotions to read, these corpus can be used to compare the analysis of different emotional state of the acoustic and rhythmic performance.

## 5.2 Evaluation index

**5.2.1 Confusion matrix.** Confusion matrix is a table, which is often used in data science and machine learning to summarize the prediction results of classification models [23]. It is represented by a matrix of $n$ rows and $n$ columns. Taking a binary classification task as an example, the structure of the confusion matrix is shown in Table 4. True Positive (TP) is the number of samples that are actually positive and predicted to be positive. False Positive (FP) is the number of samples that are actually negative but predicted to be positive. False Negative (FN) is the number of samples that are actually positive but predicted to be negative. True Negative (TN) is the number of samples that are actually negative and predicted to be negative.

In the multi classification task, confusion matrix is also applicable. The following describes the use of confusion matrix in three classification tasks, as shown in Table 5. The three categories are represented by Category 1, Category 2 and Category 3 respectively. $TP_i$ is the number of predicted correctly samples for $i$ category. $TN_{ij}$ is the number of samples for identifying category $i$ as category $j$, where $i, j \in [1, 2, 3]$. In addition, the calculation methods of TP, TN, FP and FN for each category are given in Table 5.

**Table 4. The structure of the confusion matrix.**

| Confusion matrix | | Actual value | |
|---|---|---|---|
| | | **Position** | **Negative** |
| Prdictive value | Position | TP | FP |
| | Negative | FN | TN |

**Table 5. Confusion matrix in three classification tasks.**

| Confusion matrix | | Actual value | | |
|---|---|---|---|---|
| | | **Category 1** | **Category 2** | **Category 3** |
| Predictive value | Category 1 | $TP_1$ | $TN_{12}$ | $TN_{13}$ |
| | | $TP = TP_1$ | $FP = TN_{21}+TN_{31}$ | |
| | | $FN = TP_2+TN_{23}+TN_{32}+TP_3$ | $TN = TN_{12}+TN_{13}$ | |
| | Category 2 | $TN_{21}$ | $TP_2$ | $TN_{23}$ |
| | | $TP = TP_2$ | $FP = TN_{12}+TN_{32}$ | |
| | | $FN = TP_1+TN_{13}+TN_{31}+TP_3$ | $TN = TN_{21}+TN_{23}$ | |
| | Category 3 | $TN_{31}$ | $TN_{32}$ | $TP_3$ |
| | | $TP = TP_3$ | $FP = TN_{13}+TN_{23}$ | |
| | | $FN = TP_1+TN12+TN21+TP_2$ | $TN = TN_{31}+TN_{32}$ | |

**5.2.2 Accuracy.** The accuracy rate is the proportion of correctly classified samples to the total number of samples, which is expressed as follows:

$$Accuracy = \frac{n}{N} \tag{9}$$

Where, $n$ is the number of samples that are correctly classified, and $N$ is the total number of samples. Combined with the confusion matrix in Table 5, the formula can also be written like this:

$$Accuracy = \frac{TP_1 + TP_2 + TP_3}{TP_1 + TP_2 + TP_3 + TN_{12} + TN_{13} + TN_{21} + TN_{23} + TN_{31} + TN_{32}} \tag{10}$$

## 5.3 Test the speech emotion expression ability of different feature sets

Keeping other conditions unchanged, different feature sets in Table 2 are used for speech emotion recognition. The recognition results are shown in Fig 2. The average recognition accuracy of STTF is only 0.6443, which indicates that STTF is not good at expressing speech emotion. The average expression ability of single cesptrum feature is 0.8006, and among the single cesptrum features, as the highest average recognition accuracy, NGCCs reaches 0.84, which shows that the single cesptrum features has a good emotion expression ability, but there is still a room for improvement. The average recognition rate of CESP reaches 0.9233, which shows that feature combination is conducive to the complementarity between single cesptrum features and improves the speech emotion expression ability. The average recognition rate of STTFCESP reaches 0.9461, which indicates that the combination of STTF and CESP offset the lack of information in time-domain of CESP, and reflects the advantages of feature combination and improves the speech emotion recognition accuracy.

With the intention of explaining further how feature combination complement the expression advantages of STTF and CESP, the confusion matrix of CESP and STTFCESP in Fig 3 is compared and analyzed. The differences between CESP and STTFCESP are mainly reflected in happiness, sadness and surprise categories. The accuracy of STTFCESP is 7.27%, 11.32%

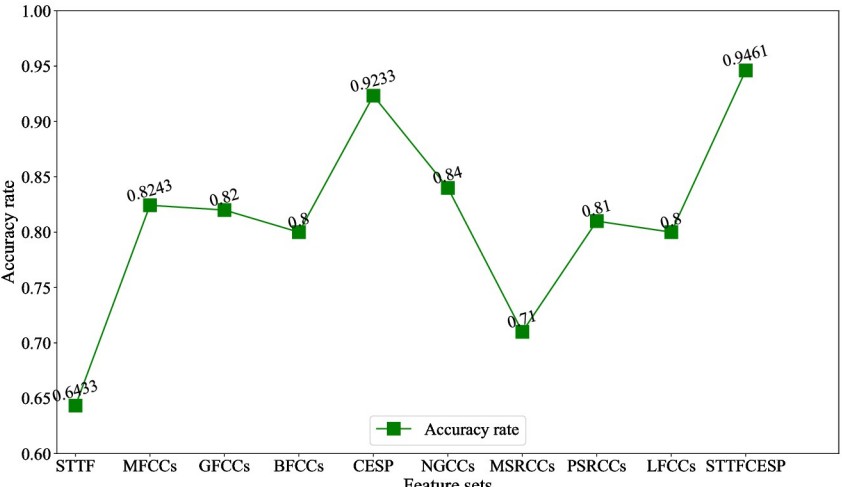

**Fig 2. The recognition results of different feature sets.**

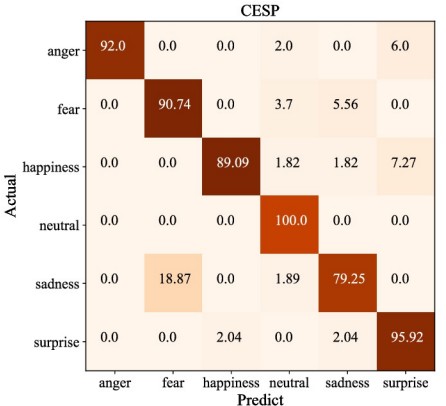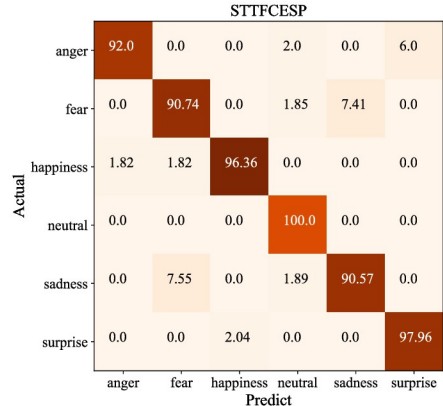

**Fig 3. The confusion matrix of CESP and STTFCESP.**

and 2.04% higher than CESP in these three categories respectively. The specific reasons are as follows:

1. In the happiness category, CESP is easy to classify it into surprise, and the recognition rate reaches 7.27%. STTFCESP eliminates this false recognition, so as to improve the expression ability of happiness category, and bring the recognition rate up to 96.36%.

2. In the sadness category, CESP is easy to classify it into fear, and the recognition rate reaches 18.87%, while STTFCESP reduces the recognition rate to 7.55%, which improves the expression ability of sadness category, and the recognition rate reaches 90.57%.

3. In the surprise category, CESP is easy to classify it into happiness and sadness, and the recognition rate is 2.04%. STTFCESP eliminate the false classification, and improve the expression ability of surprise category, and the recognition rate reaches 97.96%.

This paper draws such a conclusion that, STTF offset the lack of expression of CESP in happiness, sadness and surprise categories, and improve the overall recognition rate.

## 5.4 Verify the effectiveness of L-SFS

In the process of verifying the effectiveness of feature selection method named L-SFS, STTFCESP is used as the input of classification model, and the law that the accuracy changes with feature dimension is obtained, as shown in Fig 4. The recognition accuracy increases with the increase of feature dimension in L-SFS. When the feature dimension is 297, the model achieves the best classification performance that the average classification accuracy reaches 94.61%. Without dimensionality reduction, the average accuracy of classification is 85.65%. This demonstrates that L-SFS is effectiveness of improving the accuracy of model classification.

Experiments are conducted on the same data set and classification model to verify the effectiveness of L-SFS by comparing different feature selection methods. The experimental is designed as shown in Table 6.

In this experiment, the best expressive emotion feature set namely STTFCESP and the Sklex method are defined as the same feature set and classification model, and the feature selection methods includes LDA, Principal Component Analysis (PCA), Independent

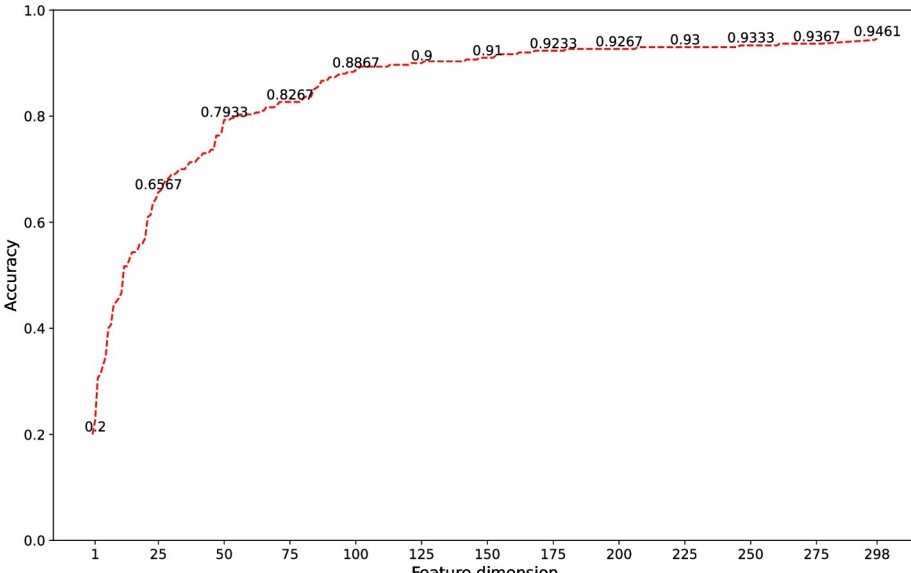

**Fig 4. The accuracy with change of feature dimension.**

Component Correlation Algorithm (ICA), L-SFS and None. Among the feature selection methods, None represents no feature dimension reduction. The classification results are shown in Fig 5.

From Fig 5, it can be seen that the average recognition accuracy of None, LDA, PCA, ICA, L-SFS are 86.56%, 90.56%, 88.65%, 87.62%, 94.61% respectively. Therefore, the following two conclusions are drawn.

1. After the step of feature selection, the recognition accuracy is boosted evidently. It shows that feature selection is helpful to improve the recognition accuracy.

2. In those feature selection methods, the feature set after L-SFS has stronger expression ability and the recognition accuracy is the highest, which verifies the effectiveness of L-SFS.

## 5.5 Verify the effectiveness of learned automatically weights

Facing the defect of the ensemble learning of voting strategy, this paper uses the softmax regression model to learn automatically weights. In order to verify the effectiveness of this method, three randomly selected weight vectors, $w1$, $w2$, and $w3$ are compared with the weight vector $w4$ learned by the softmax regression model. The experimental design is shown in Table 7.

**Table 6. The feature selection methods.**

| Feature set | Feature selection methods | Classifier |
|---|---|---|
| STTFCESP | LDA | Sklex |
| | PCA | |
| | ICA | |
| | L-SFS | |
| | None | |

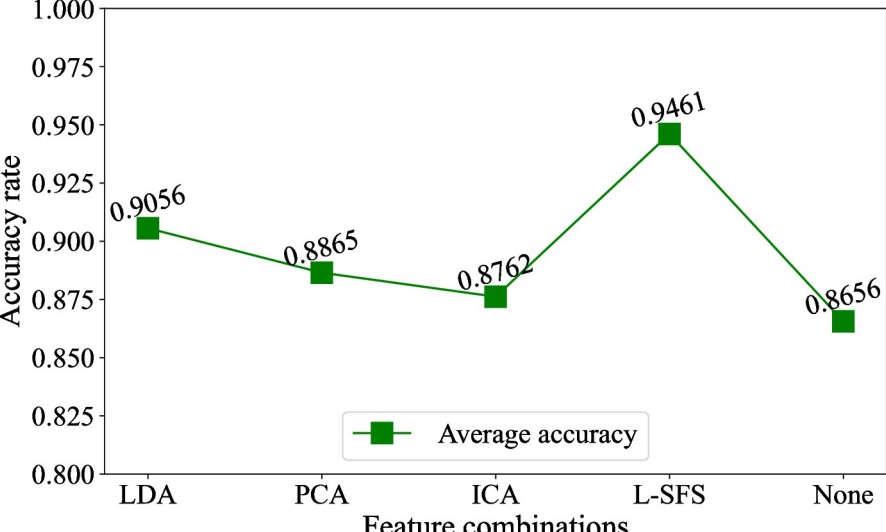

**Fig 5. The classification results.**

The experimental results are shown in the Fig 6. The recognition accuracy of each category fluctuates greatly indicates that the different weights have a great impact on the results of the ensemble learning of voting strategy, and scientific weights can optimize the ensemble learning of voting strategy. In addition, the learned automatically weights perform best in the classification results that the accuracy reaches 94.6%, which shows the effectiveness of this method.

## 5.6 Test the speech emotion recognition ability of Sklex method

This paper uses ensemble learning method to solve the speech emotion recognition task. In order to verify the effectiveness of Sklex method, the experiment is designed as it shown in Table 8.

In order to select the optimal parameters for the four weak classifiers in the integration strategy, this paper optimizes the parameters of the weak classifiers respectively based on the control variate method, as shown in the Table 9. When tuning the hyperparameters of SVM, the other three classifiers are initialized. The tuned SVM and the other two initialized classifiers are used to tune the hyperparameters of XGBoost. The tuned SVM and XGBoost, and the initialized KNN are used to tune the hyperparameters of LightGBM. When tuning the parameters of KNN, the tuned SVM, XGBoost and LightGBM are used.

In this paper, the grid method is used to tune hyperparameters, and the hyperparameters optimization range is set for SVM, XGBoost, LightGBM and KNN respectively, as shown in

**Table 7. The weight vectors assigned.**

| Feature set | Weight vectors | Classifier |
|---|---|---|
| STTFCESP | $w1 = [1, 1, 1, 1]$ | Sklex |
| | $w2 = [3, 1, 2, 2]$ | |
| | $w3 = [3, 1, 3, 2]$ | |
| | $w4 = [6, 2, 3, 3]$ | |

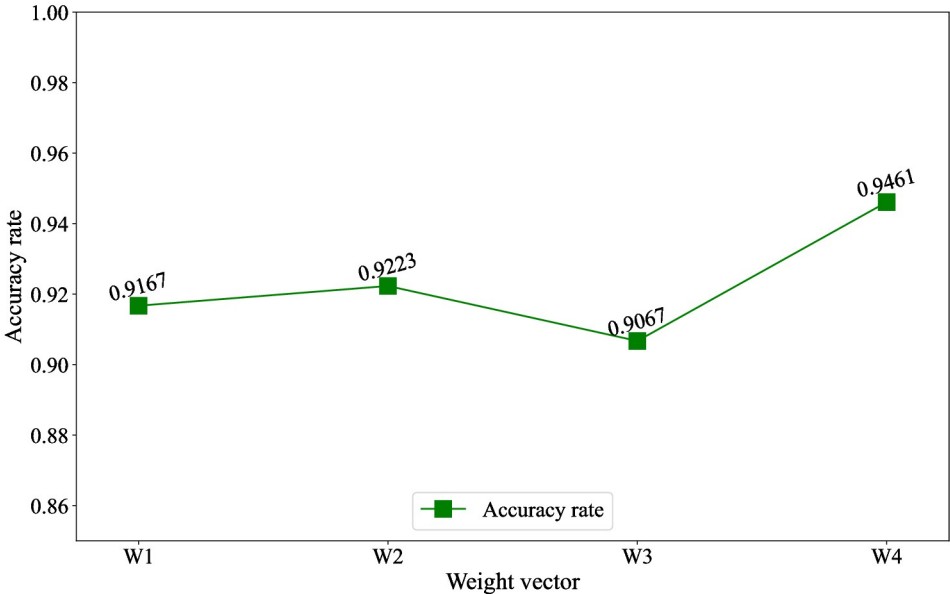

**Fig 6. The recognition results.**

**Table 8. The classfication methods.**

| Feature set | Classfier | | | | |
|---|---|---|---|---|---|
| STTFCESP | SVM | XGBoost | LightGBM | KNN | Sklex |

**Table 9. Classifier parameter tuning.**

| Tuning classifier | classifier | | | |
|---|---|---|---|---|
| | **SVM** | **XGBoost** | **LightGBM** | **KNN** |
| SVM | **tuning** | initialize | initialize | initialize |
| XGBoost | tuned | **tuning** | initialize | initialize |
| LightGBM | tuned | tuned | **tuning** | initialize |
| KNN | tuned | tuned | tuned | **tuning** |

**Table 10. Optimization range of SVM parameters.**

| probability | C | gamma | kernel |
|---|---|---|---|
| True | [1, 2, 5, 10] | [0.01, 0.05, 0.1, 0.3, 0.5, 1] | ['linear', 'rbf'] |

the Tables 10–13. And then the four weak classifiers are trained to obtain the super parameters, as shown in Tables 14–17, and the four classifiers are integrated.

In this paper, Sklex and four single weak learners including SVM, XGBoost, LightGBM and KNN, are compared through experiments. The effectiveness and stability of the Sklex method are verified by comparing the classification results of Sklex and the four single learners. After the experiment, the results are shown in Fig 7.

**Table 11. Optimization range of XGBoost parameters.**

| verbosity | random_state | learning rate |
|-----------|--------------|---------------|
| True | 0 | [0.3, 0.5, 0.7] |
| **max depth** | **n estimators** | **objective** |
| [3, 4, 5, 6] | [50, 100, 200] | ['multi:softmax', 'multi:softprob'] |

**Table 12. Optimization range of LightGBM parameters.**

| num leaves | objective | boosting type |
|------------|-----------|---------------|
| True | [1, 2, 5, 10] | [gbdt, dart, goss] |
| **learning rate** | **max depth** | **n estimators** |
| [0.01, 0.1, 1] | [16, 17, 18, 20, 22] | [20, 40, 100] |

**Table 13. Optimization range of KNN parameters.**

| algorithm | leaf size | p |
|-----------|-----------|---|
| ['auto', 'ball tree', 'kd tree', 'brute'] | [5, 10, 20, 30, 50, 100] | [1, 2, 3] |
| **weights** | **n neighbors** | |
| ['uniform', 'distance'] | [1, 5, 10, 15, 20, 50, 100] | |

**Table 14. Super parameters of SVM.**

| probability | C | gamma | kernel |
|-------------|---|-------|--------|
| True | 10 | 0.05 | rbf |

**Table 15. Super parameters of XGBoost.**

| verbosity | random state | learning rate |
|-----------|--------------|---------------|
| 1 | 0 | 0.3 |
| max depth | n estimators | objective |
| 5 | 200 | multi:softmax |

**Table 16. Super parameters of LightGBM.**

| num leaves | objective | boosting type | learning rate | max depth | n estimators |
|------------|-----------|---------------|---------------|-----------|--------------|
| 10 | multiclass | gbdt | 0.1 | 16 | 100 |

It can be seen from Fig 7, the average recognition accuracy of SVM, XGBoost, LightGBM, KNN and Sklex are 92.78%, 86.12%, 87.39%, 88.09%, 92.78% and 94.6% respectively. The average recognition accuracy of Sklex is the highest, which shows that the ensemble method is effective to improving the classification accuracy, and the effectiveness of Sklex method is verified.

**Table 17. Super parameters of KNN.**

| algorithm | leaf size | n neighbors | p | weights |
|-----------|-----------|-------------|---|---------|
| auto | 5 | 20 | 1 | uniform |

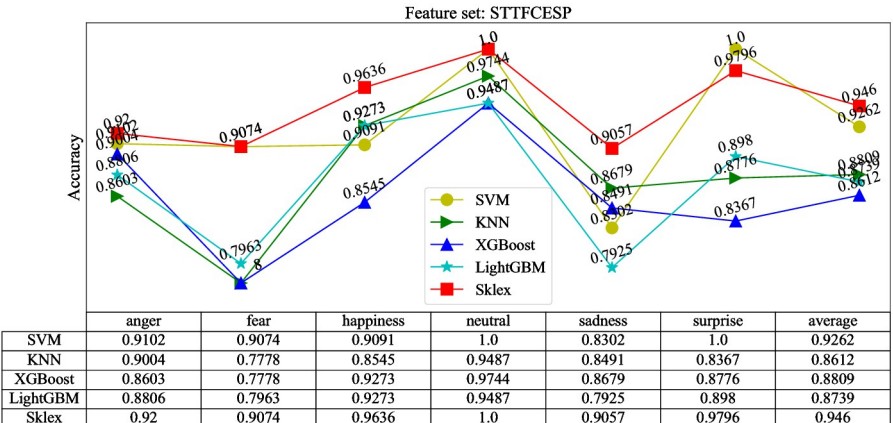

| | anger | fear | happiness | neutral | sadness | surprise | average |
|---|---|---|---|---|---|---|---|
| SVM | 0.9102 | 0.9074 | 0.9091 | 1.0 | 0.8302 | 1.0 | 0.9262 |
| KNN | 0.9004 | 0.7778 | 0.8545 | 0.9487 | 0.8491 | 0.8367 | 0.8612 |
| XGBoost | 0.8603 | 0.7778 | 0.9273 | 0.9744 | 0.8679 | 0.8776 | 0.8809 |
| LightGBM | 0.8806 | 0.7963 | 0.9273 | 0.9487 | 0.7925 | 0.898 | 0.8739 |
| Sklex | 0.92 | 0.9074 | 0.9636 | 1.0 | 0.9057 | 0.9796 | 0.946 |

**Fig 7. The recognition results of learner.**

With the intention of further studying the reason why Sklex improves the accuracy, the confusion matrices of SVM and Sklex are analyzed in this section. The confusion matrix is shown in the Fig 8. The reason why Sklex improves the recognition accuracy is that it improves the ability to distinguish every single category. Compared with SVM, the recognition accuracy of Sklex in happiness and sadness is improved by 5.54% and 7.74% respectively. The reason is that Sklex avoids the false identification of happiness as neutral, sadness and surprise, and reduces the probability of false identification of sadness as fear.

For the purpose of verifying the effectiveness of Sklex method, the results of this method and other methods are compared on the CASIA database, and the results are shown in the Table 18.

From Table 18, the average recognition rate of Sklex is higher than other methods, which is mainly manifested in four emotions including fear, happiness, neutral and surprise. And it is slightly lower than literature [2] in sad emotion. In general, this method has achieved ideal results in speech emotion recognition.

## 6 Conclusion and prospect

In order to improve the level of Human-computer interaction, a novel speech emotion recognition method based on feature construction and ensemble learning is proposed in this paper.

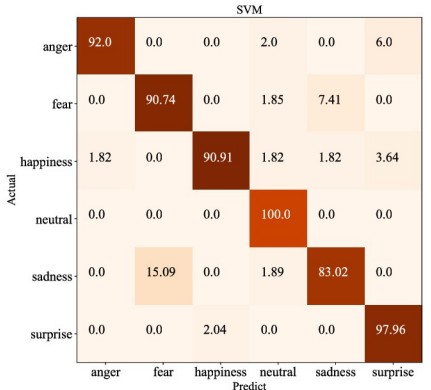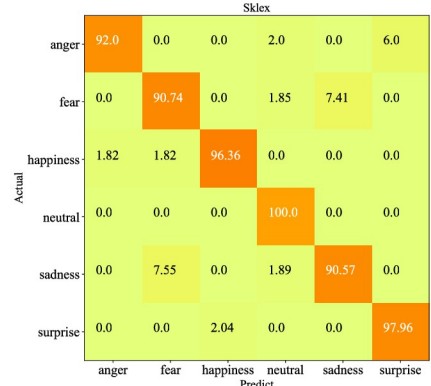

**Fig 8. The confusion matrices of SVM and Sklex.**

**Table 18. Comparison of results.**

| Methods | Accuracy | | | | | | |
|---------|----------|------|-----------|---------|---------|----------|---------|
| | anger | fear | happiness | neutral | sadness | surprise | average |
| [24] | 0.86 | 0.85 | 0.75 | 0.93 | 0.77 | 0.89 | 0.8417 |
| [3] | 0.92 | 0.79 | 0.84 | 0.94 | 0.86 | 0.96 | 0.885 |
| [2] | 0.92 | 0.84 | 0.90 | 0.95 | 0.91 | 0.91 | 0.905 |
| Sklex | 0.92 | 0.9074 | 0.9636 | 1.00 | 0.9057 | 0.9796 | 0.946 |

In the CASIA database, the experimental results show that the recognition rates of anger, fear, happiness, neutral, sadness and surprise are 92.00%, 90.74%, 96.36%, 100.00%, 90.57% and 97.96% respectively. In conclusion, the method reflects the effectiveness of feature construction and the superiority and stability of ensemble learning.

In this paper, the traditional acoustic feature extraction process reflects the human diligence and wisdom, but there is still not a complete feature set. Besides, the ensemble learning method achieves good results, but it is not clear that increasing (reducing) the number of single weak learner and changing the type of learner will improve the recognition accuracy. So it remains be further tested.

With the development of deep learning, researchers use the method based on autoencoder to extract automatically acoustic features. This method is conducive to the construction of a complete feature set, and may find some new features. Therefore, in the future, we can start from the way of automatic learning features to build a feature set with stronger speech emotion expression ability.

## Supporting information

**S1 File.**
(PDF)

## Acknowledgments

In the process of research, I encountered many difficulties and learned a lot of knowledge and skills. First of all, I would like to thank my instructor, Mr. Guo Yi. In the process of basic theoretical knowledge and algorithm practice, Mr. Guo has given me a lot of guidance on these problems and given me valuable reference direction. In the process of writing the paper, Mr. Guo guided me from a professional point of view, helped me to sort out the framework of my paper, and gave a detailed explanation of each content. Secondly, I would like to thank my graduate friends for your company and support. Finally, I would like to thank the editors and judges for their hard work.

## Author Contributions

**Conceptualization:** Yi Guo.

**Data curation:** Xuejun Xiong.

**Methodology:** Yi Guo, Xuejun Xiong.

**Resources:** Xuejun Xiong.

**Software:** Xuejun Xiong, Yangcheng Liu.

**Supervision:** Yi Guo.

**Validation:** Yangcheng Liu, Qiong Li.

**Writing – original draft:** Xuejun Xiong.

**Writing – review & editing:** Xuejun Xiong, Liang Xu, Qiong Li.

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
