## [Decision Letter · Decision Letter 0]

15 Dec 2021

PONE-D-21-34736A novel speech emotion recognition method based on feature construction and ensemble learningPLOS ONE

Dear Dr. Xuejun,

Thank you for submitting your manuscript to PLOS ONE. After careful consideration, we feel that it has merit but does not fully meet PLOS ONE’s publication criteria as it currently stands. Therefore, we invite you to submit a revised version of the manuscript that addresses the points raised during the review process.

We look forward to receiving your revised manuscript.

Kind regards,

Sathishkumar V E

Academic Editor

PLOS ONE

Journal Requirements:

 (This research was supported by the Open Research Fund of the Power System Wide-area Measurement and Control Sichuan Provincial Key Laboratory (2020JDS0027), Sichuan Science & Technology Program (2020YJ0367, 2021YFG0337), the National Natural Science Foundation of China under Grant (61973257).  The funders had no role in study design, data collection and analysis, decision to publish, or preparation of the manuscript.)

6. Please include a caption for figure 4, and 5. 

8. We suggest you thoroughly copyedit your manuscript for language usage, spelling, and grammar. If you do not know anyone who can help you do this, you may wish to consider employing a professional scientific editing service. 

Reviewers' comments:

Reviewer's Responses to Questions

**Comments to the Author**

1. Is the manuscript technically sound, and do the data support the conclusions?

Reviewer #1: Partly

Reviewer #2: Yes

2. Has the statistical analysis been performed appropriately and rigorously? 

Reviewer #1: Yes

Reviewer #2: No

3. Have the authors made all data underlying the findings in their manuscript fully available?

Reviewer #1: Yes

Reviewer #2: Yes

4. Is the manuscript presented in an intelligible fashion and written in standard English?

Reviewer #1: Yes

Reviewer #2: Yes

5. Review Comments to the Author

Reviewer #1: In this article, the authors propose novel speech emotion recognition method based on feature construction and ensemble learning to improve the level of Human-computer interaction. The article seems novel and the contributions are considerable for the publication. However, there are several corrections required before consider this paper for publication in this journal.

~ The literature study of this article is poor. It is recommended to consider the recent literautre and also provide the justification that how the proposed work is better over the existing ones.

~ The feature section (Section 3) need more detailed discussion.

~ Provide the citations for the datasets used in this work.

~ The time complexity for the proposed algorithms to be estimated and compared using the existing models.

~ There are several machine learning based classification algorithm but the authors studied very few in this paper. Why authors considered only few. Provide the justifications or refer. Machine learning algorithms for wireless senor networks: a survey for comparisons on various classification algorithms.

~ Compare the model using recent existing algorithms.

~ list the limitations on proposed work.

Reviewer #2: The article presented a novel speech emotion recognition system with feature extraction and selection. The work showed good results. The foloowing are the points to be addressed by the authors for further processing of the article.

1. How does L-SFS help to extract and select features? This is missing.

2. The datast CASIA needs to be explored.

3. The rationale for choosing SVM, KNN, LightGBM and XGBoost for constructing proposed Sklex model

4. The research questions to be addressed by the proposed work and research motivation need to be strengthed.

5. Why ensembling of four models give best results?

6. What are the hyper-parameters used for the experiment?

6. PLOS authors have the option to publish the peer review history of their article (what does this mean?). If published, this will include your full peer review and any attached files.

Reviewer #1: No

Reviewer #2: No

---

## [Author Response · Author response to Decision Letter 0]

27 Feb 2022

This letter that responds to each point raised by the academic editor and the two reviewers. Firstly, I thank the editorial department very much for its recognition of this research work and detailed modification opinions. Secondly, I am very grateful to the two reviewers for their hard work and detailed review of the report. All of the points raised by editor and reviewers are very important for me. So, following the points, we carefully revise this paper and provide some responses as follows.

 Response to the academic editor. By carefully reading the modification suggestions and schemes put forward by the academic editor, we have revised the structure and submission process of the paper, which can be reflected in the new submission.

Response to the reviewer 1.

1.The literature study of this paper is poor. It is recommended to consider the recent literautre and also provide the justification that how the proposed work is better over the existing ones.

 In this paper, we propose a method to improve the level of human-computer interaction. The main purpose of this method is to improve the performance of speech emotion recognition. The references in this paper are not up-to-date, but conform to the research theme of this paper. At the same time, the experimental comparison of this paper is carried out for the data set CASIA (Chinese emotion corpus).

 According to the revision point, in CSCD and SCI database we searched the recent research of speech emotion recognition based on CASIA data set. The latest research literature are mostly based on neural network, and the model complexity is high and the performance is not very good. For example, the accuracy of a novel heterogeneous parallel revolution Bi LSTM for speech emotion recognition in CASIA data set is only 79.67%, the accuracy rate of a novel user emotional interaction design model using long and short term memory networks and deep learning on CASIA data set is only 72.5%, and the accuracy rate of speech emotion recognition based on transfer learning from the facenet framework (a) on CASIA data set is only 90%, the accuracy of attention based revolution skip bidirectional long short term memory network for speech emotion recognition in CASIA data set has only reached 72.5%. Therefore, we still insist on the effectiveness of the original references.

 2.The feature section (Section 3) need more detailed discussion.

 In the section 3 (feature construction), firstly we introduced how to extract speech features (feature extraction), and then we introduced how to select subset from feature set (feature selection). 

 In feature extraction subsection, we have referred to many literatures on speech signal processing, after experimental comparison (not appearing in the paper), we select the acoustic features in the paper and calculate the statistical characteristics of these acoustic features. How to calculate these acoustic features and their statistical characteristics are not detail discussed in the paper, because these calculation processes can be found in references.

 In feature selection subsection, we mainly introduced the feature selection method of L-SFS, and the detailed calculation steps of this step are described in detail in this paper.

3.Provide the citations for the datasets used in this work.

 Following the revision point, we add “CASIA Chinese emotional corpus was recorded by the Institute of Automation, Chinese Academy of Sciences. It includes four professional speakers and six kinds of emotions: anger, fear, happiness, neutral, sadness and surprise, a total of 7200 different pronunciation. 300 of the corpus are the same text. That is to say the same text given different emotions to read, these corpus can be used to compare the analysis of different emotional state of the acoustic and rhythmic performance.” to the paper. 

4.The time complexity for the proposed algorithms to be estimated and compared using the existing models.

 The main research of this paper is to improve the performance of speech emotion recognition. There is no quantitative analysis of the time complexity of the model, but from the perspective of qualitative analysis, the time complexity of the model in this paper is more complex than a single classifier and simpler than the classifier based on neural network. 

5.There are several machine learning based classification algorithm but the authors studied very few in this paper. Why authors considered only few. Provide the justifications or refer. Machine learning algorithms for wireless senor networks: a survey for comparisons on various classification algorithms.

 We know that there are many classification models based on machine learning algorithms , such as linear regression, Bayesian classifier, decision tree, random forest, k-nearest neighbor, support vector machine, etc., but this paper selects k-nearest neighbor, support vector machine, xgboost and lightgbm. The reason is that after reading a lot of references, these four classifiers perform well in speech emotion classification, and the other classifiers do not perform well.

6.Compare the model using recent existing algorithms.

 As the answer to point 1, most of the latest research is based on neural network, and its performance on CASIA data set is not as good as the method proposed in this paper. So, we decided not to add the modification of this part in this paper.

7.List the limitations on proposed work.

1)The research content proposed in this paper is for CASIA data set, which is for Chinese speech emotion recognition. So, there is a limitation of cross language emotional expression.

2)In this paper, the traditional acoustic feature extraction process reflects the human diligence and wisdom, but there is still not a complete feature set. Besides, the ensemble learning method achieves good results, but it is not clear that increasing (reducing) the number of single weak learner and changing the type of learner will improve the recognition accuracy. So, it remains be further tested.

Response to the reviewer 2.

1.How does L-SFS help to extract and select features? This is missing.

 In the feature selection subsection, we have introduced the steps of L-SFS feature selection method in detail. In addition, in subsection 5.4, we have verified this method in the form of experiments. 

2.The dataset CASIA needs to be explored.

 Following the revision point, we add “CASIA Chinese emotional corpus was recorded by the Institute of Automation, Chinese Academy of Sciences. It includes four professional speakers and six kinds of emotions: anger, fear, happiness, neutral, sadness and surprise, a total of 7200 different pronunciation. 300 of the corpus are the same text. That is to say the same text given different emotions to read, these corpus can be used to compare the analysis of different emotional state of the acoustic and rhythmic performance.” to the paper. 

3.The rationale for choosing SVM, KNN, LightGBM and XGBoost for constructing proposed Sklex model.

 We know that there are many classification models based on machine learning algorithms , such as linear regression, Bayesian classifier, decision tree, random forest, k-nearest neighbor, support vector machine, etc., but this paper selects k-nearest neighbor, support vector machine, xgboost and lightgbm. The reason is that after reading a lot of references, these four classifiers perform well in speech emotion classification, and the other classifiers do not perform well.

4.The research questions to be addressed by the proposed work and research motivation need to be strengthed.

5.Why ensembling of four models give best results?

 Actually, we are not sure that selecting four classifiers will achieve the best performance. At present, the classification effect is ideal. If more classifiers are added, it may cause model flooding and increase the time complexity of the model; At the same time, reducing the classifier will reduce the performance of the model.

 In the paper of Conclusion and Prospect section, we mentioned that “the ensemble learning method achieves good results, but it is not clear that increasing (reducing) the number of single weak learner and changing the type of learner will improve the recognition accuracy. So it remains be further tested.”, so we are not sure that choosing four classifiers will achieve the best performance.

6.What are the hyper-parameters used for the experiment?

Thank you very much for raising this question. We have made changes in the article, and the results are as follow tables.

---

## [Decision Letter · Decision Letter 1]

28 Mar 2022

PONE-D-21-34736R1A novel speech emotion recognition method based on feature construction and ensemble learningPLOS ONE

Dear Dr. Xuejun,

Thank you for submitting your manuscript to PLOS ONE. After careful consideration, we feel that it has merit but does not fully meet PLOS ONE’s publication criteria as it currently stands. Therefore, we invite you to submit a revised version of the manuscript that addresses the points raised during the review process.

We look forward to receiving your revised manuscript.

Kind regards,

Sathishkumar V E

Academic Editor

PLOS ONE

Reviewers' comments:

Reviewer's Responses to Questions

**Comments to the Author**

1. If the authors have adequately addressed your comments raised in a previous round of review and you feel that this manuscript is now acceptable for publication, you may indicate that here to bypass the “Comments to the Author” section, enter your conflict of interest statement in the “Confidential to Editor” section, and submit your "Accept" recommendation.

Reviewer #1: All comments have been addressed

Reviewer #2: (No Response)

2. Is the manuscript technically sound, and do the data support the conclusions?

Reviewer #1: Partly

Reviewer #2: Yes

3. Has the statistical analysis been performed appropriately and rigorously? 

Reviewer #1: Yes

Reviewer #2: No

4. Have the authors made all data underlying the findings in their manuscript fully available?

Reviewer #1: No

Reviewer #2: Yes

5. Is the manuscript presented in an intelligible fashion and written in standard English?

Reviewer #1: Yes

Reviewer #2: Yes

6. Review Comments to the Author

Reviewer #1: The authors are addressed all the recommended comments and the current version is well improved over the previous version of the article. So, it is recommended for publication in this journal.

Reviewer #2: The authors have not listed the hyperparameters and their tuning approaches.

The authors have not mentioned the research focus and questions to be addressed by the research work.

The response to Question 5 raised during review 1 has not been clearly addressed. The authors themselves are not sure about the performance of the work. Intensive experiments to be conducted.

7. PLOS authors have the option to publish the peer review history of their article (what does this mean?). If published, this will include your full peer review and any attached files.

Reviewer #1: No

Reviewer #2: No

---

## [Author Response · Author response to Decision Letter 1]

29 Mar 2022

Response to Reviewers

 This letter that responds to each point raised by the academic editor and the two reviewers. Firstly, I thank the editorial department very much for its recognition of this research work and detailed modification opinions. Secondly, I am very grateful to the two reviewers for their hard work and detailed review of the report. All of the points raised by editor and reviewers are very important for me. So, following the points, we carefully revise this paper and provide some responses as follows.

Response to the academic editor. 

 By carefully reading the modification suggestions and schemes put forward by the academic editor, we have revised the structure and submission process of the paper, which can be reflected in the new submission.

Response to the reviewer 1.

 Thank you for your affirmation of the revised paper, which has had a positive impact on our research work.

Response to the reviewer 2.

1. The authors have not listed the hyperparameters and their tuning approaches.

 In fact, in the revised version, we have given the hyperparameters of the weak classifier, but we did not give the tuning method of the hyperparameters. Therefore, in the new revised version, we give the tuning method and the range setting of the super parameters before tuning. As shown in subsection 5.6 in this paper, and shown below:

2. The authors have not mentioned the research focus and questions to be addressed by the research work. 

 Based on this point, this paper mainly studies a Chinese speech emotion recognition method based on feature combination and ensemble learning, in order to improve the recognition accuracy of Chinese speech emotion. Because affective features not well represent emotions, we use feature combination method to obtain a new speech affective feature set. To solve the problem of low recognition accuracy of weak classifiers, we take an ensemble learning strategy, and use Softmax to automatically learn the weight of weak classifiers. The experimental results show that the new speech emotion feature set has good representation ability, and the ensemble learning idea also improves the recognition accuracy. This is mentioned in the abstract, introduction and experimental analysis of this paper.

3. The response to Question 5 raised during review 1 has not been clearly addressed. The authors themselves are not sure about the performance of the work. Intensive experiments to be conducted.

 I'm sorry for the last reply and we have made new adjustments. In the last reply, we mentioned that we are not sure that the integration effect of these four classifiers is the best, but experiments show that they have indeed achieved good performance on CASIA data sets. Therefore, adding other classifiers for integration is not the focus of this paper.

---

## [Decision Letter · Decision Letter 2]

4 Apr 2022

A novel speech emotion recognition method based on feature construction and ensemble learning

PONE-D-21-34736R2

Dear Dr. Xuejun,

We’re pleased to inform you that your manuscript has been judged scientifically suitable for publication and will be formally accepted for publication once it meets all outstanding technical requirements.

Kind regards,

Sathishkumar V E

Academic Editor

PLOS ONE

Additional Editor Comments (optional):

Reviewers' comments:

Reviewer's Responses to Questions

**Comments to the Author**

1. If the authors have adequately addressed your comments raised in a previous round of review and you feel that this manuscript is now acceptable for publication, you may indicate that here to bypass the “Comments to the Author” section, enter your conflict of interest statement in the “Confidential to Editor” section, and submit your "Accept" recommendation.

Reviewer #1: All comments have been addressed

Reviewer #2: All comments have been addressed

2. Is the manuscript technically sound, and do the data support the conclusions?

Reviewer #1: Partly

Reviewer #2: Yes

3. Has the statistical analysis been performed appropriately and rigorously? 

Reviewer #1: Yes

Reviewer #2: Yes

4. Have the authors made all data underlying the findings in their manuscript fully available?

Reviewer #1: No

Reviewer #2: Yes

5. Is the manuscript presented in an intelligible fashion and written in standard English?

Reviewer #1: No

Reviewer #2: Yes

6. Review Comments to the Author

Reviewer #1: The authors addressed all the recommended comments and the current version is recommended for publication in this journal.

Reviewer #2: The authors have addressed all the queries raised during revision 1 and 2. For the last query raised during revision 2, the authors have mentioned that the work will be considered in near future.

7. PLOS authors have the option to publish the peer review history of their article (what does this mean?). If published, this will include your full peer review and any attached files.

Reviewer #1: No

Reviewer #2: No

---

## [Editor Report · Acceptance letter]

19 Apr 2022

PONE-D-21-34736R2 

A novel speech emotion recognition method based on feature construction and ensemble learning 

Dear Dr. Xuejun:

I'm pleased to inform you that your manuscript has been deemed suitable for publication in PLOS ONE. Congratulations! Your manuscript is now with our production department. 

Kind regards, 

on behalf of

Dr. Sathishkumar V E 

Academic Editor

PLOS ONE